# Research Progress in Diffusion Spectrum Imaging

**DOI:** 10.3390/brainsci13101497

**Published:** 2023-10-23

**Authors:** Fenfen Sun, Yingwen Huang, Jingru Wang, Wenjun Hong, Zhiyong Zhao

**Affiliations:** 1Center for Brain, Mind and Education, Shaoxing University, Shaoxing 312000, China; ffsun_psy2020@foxmail.com (F.S.); yingwenhuang2023@foxmail.com (Y.H.); fwxs15531818790@163.com (J.W.); 2Department of Rehabilitation Medicine, Afiliated Drum Tower Hospital, Medical School of Nanjing University, Nanjing 210008, China; hongwenjun1989@126.com; 3Key Laboratory for Biomedical Engineering of Ministry of Education, Department of Biomedical Engineering, College of Biomedical Engineering & Instrument Science, Zhejiang University, Hangzhou 310027, China

**Keywords:** diffusion magnetic resonance imaging, diffusion spectrum imaging, progress, methodology, application

## Abstract

Studies have demonstrated that many regions in the human brain include multidirectional fiber tracts, in which the diffusion of water molecules within image voxels does not follow a Gaussian distribution. Therefore, the conventional diffusion tensor imaging (DTI) that hypothesizes a single fiber orientation within a voxel is intrinsically incapable of revealing the complex microstructures of brain tissues. Diffusion spectrum imaging (DSI) employs a pulse sequence with different b-values along multiple gradient directions to sample the diffusion information of water molecules in the entire q-space and then quantitatively estimates the diffusion profile using a probability density function with a high angular resolution. Studies have suggested that DSI can reliably observe the multidirectional fibers within each voxel and allow fiber tracking along different directions, which can improve fiber reconstruction reflecting the true but complicated brain structures that were not observed in the previous DTI studies. Moreover, with increasing angular resolution, DSI is able to reveal new neuroimaging biomarkers used for disease diagnosis and the prediction of disorder progression. However, so far, this method has not been used widely in clinical studies, due to its overly long scanning time and difficult post-processing. Within this context, the current paper aims to conduct a comprehensive review of DSI research, including the fundamental principles, methodology, and application progress of DSI tractography. By summarizing the DSI studies in recent years, we propose potential solutions towards the existing problem in the methodology and applications of DSI technology as follows: (1) using compressed sensing to undersample data and to reconstruct the diffusion signal may be an efficient and promising method for reducing scanning time; (2) the probability density function includes more information than the orientation distribution function, and it should be extended in application studies; and (3) large-sample study is encouraged to confirm the reliability and reproducibility of findings in clinical diseases. These findings may help deepen the understanding of the DSI method and promote its development in clinical applications.

## 1. Introduction

Diffusion magnetic resonance imaging (dMRI) is the only way to noninvasively measure structural connectivity in the human brain, and it has been extensively applied in clinical studies [1]. As one of the most commonly used dMRI techniques, diffusion tensor imaging (DTI) reconstructs structural connectivity patterns based on the phenomenon of the diffusion anisotropy of water molecules in brain tissue. In the DTI model, two metrics, the apparent diffusion coefficient (ADC) and fraction anisotropy (FA), are particularly sensitive to brain lesions in diseases [2,3], and the fiber tract is tracked by the main direction of the diffusion tensor [4]. However, in the past twenty years, an increasing number of studies have reported that the fiber architectures in the human brain are more complicated than we thought, such as crossing or branching fibers within a single voxel [5,6]. The distances revealed by dMRI are far larger than the diffusion scale, implying that each 3D resolution element (voxel) represents many distinct diffusional environments. This provides a complex diffusion profile that is generally underspecified by the six degrees of freedom of the tensor model. Therefore, DTI is incapable of resolving multiple fiber orientations within a single voxel [7]. Additionally, the FA value is lower when a multidirectional fiber exists in a single voxel, whereas the degeneration of tissue integrity also causes a reduced FA value. Thus, the FA metric may be inappropriate to evaluate the structural integrity of the tissue with crossing fibers. This not only affects the accuracy of the DTI technique in clinical diagnosis but also leads to the failure of fiber tract tracking in complex tissue.

To resolve the multiple fiber orientations within individual voxels, various methods have been proposed. They have been classified into two types: one is model-based, such as multi-tensor diffusion imaging [8], diffusion kurtosis imaging (DKI) [9], neurite orientation dispersion and density imaging (NODDI) [10], the ball and stick model [11], the persistent angular structure MRI [12], and diffusion orientation transform [12]; the other is model-free high-angular-resolution diffusion imaging (HARDI), which describes the diffusion motion by the orientation distribution function (ODF). For instance, diffusion spectrum imaging (DSI) reconstructs the ODF by applying an inverse Fourier transform (FT) to the q-space data from the grid sampling scheme [13]; Q-ball imaging (QBI) based on a Funk–Radon (FR) transform reconstructs the ODF from a single-shell dataset [14]; hybrid diffusion imaging (HYDI) concurrently performs DSI, Q-ball, and DTI analysis in multi-shell data [15]; and generalized-sampling imaging (GQI) describes the diffusion behavior using the spin distribution function (SDF), similar to ODF [16].

Wedeen et al. first proposed the DSI, a multi-b-value and multidirectional q-space imaging method, which calculates the diffusion ODF by applying the FT to the diffusion MR signals and conducting radial integration [17]. This method can successfully reveal crossing fibers. Moreover, the general fraction anisotropy (GFA) produced from DSI is more sensitive to tissue degeneration in diseased brains and has better accuracy and precision in identifying the multidirectional fibers than the conventional FA obtained using DTI [18]. Under a relatively unified standard, one prior study [19] compared 19 dMRI approaches, in which the dMRI methods were classified into four types according to the pattern of the data sampling scheme: DTI-like (single ball and lower b-value sample), HARDI-like (single ball and higher b-value sample), DSI-like (Cartesian grid sample), and Strike-like (sparse sample). The results showed that compared with other high-resolution imaging techniques, the DSI was more precise and stable than the DTI in terms of both the angular accuracy and the success rate of reconstructing the crossing fiber and the ability to uncover the minimal angle of the crossing fiber (Figure 1). However, so far, no study has comprehensively reviewed the research progress of DSI.

In this paper, we first simply describe the basic theory of several dMRI techniques. Second, we illustrate methodological studies on DSI from two perspectives: the improvement in the scanning time and the optimization of post-processing. Third, we primarily depict the progress of DSI in revealing the tissue microstructures and clinical applications. Finally, we summarize the paper and illustrate the current problems that remain for DSI. The design of the entire work is shown in Figure 2. 

## 2. Methodology

It is well known that in the classic Stejskal–Tanner experiment, the MR signal is made proportional to the voxel average dephasing for a specified diffusion duration Δ:(1)$$S_{\Delta }=S_{0}\langle e^{i\Phi }\rangle ,$$
where $S_{0}$ is a constant that can be computed by the spin-echo experiment without diffusion weighting, and $S_{\Delta }$ and Φ represent the MR signal with diffusion weighting and dephasing, respectively. 

We assume in the first instance that the duration δ of the diffusion sensitizing gradient is negligible compared to the mixing time Δ. Thus, we have $\Phi =\vec{q}.\vec{r}\left(\vec{q}=\delta \gamma \vec{g},\;\vec{r}=\vec{x}\left(\Delta \right)-\vec{x}\left(0\right)\right)$, where $\vec{x}\left(0\right)$ and $\vec{x}\left(\Delta \right)$ must be understood as the spin position at the time of the application of the first and second diffusion gradient pulses, respectively. The gradient wave vector is defined as $\vec{q}=\delta \gamma \vec{g}$, where γ is the gyromagnetic ratio, $\vec{g}$ is the gradient vector, and $\vec{r}$ is the relative spin displacement between the first diffusion gradient and second diffusion gradient.

Considering the voxel average as an expectation E(·), the MR signal is proportional to the characteristic function of the relative spin displacement vector. This yields a Fourier relationship between the MR signal and the underlying density ${\vec{p}}_{\Delta }\left(\vec{r}\right)$:(2)$$S_{\Delta }\left(\vec{q}\right)=S_{0}E\left(e^{i\Phi }\right)=S_{0}\int^{\;}{\vec{p}}_{\Delta }\left(\vec{r}\right)e^{i\vec{q}\cdot \vec{r}}d^{3}\vec{r}$$
where ${\vec{p}}_{\Delta }\left(\vec{r}\right)$ represents the density of the average relative spin displacement in a voxel. 

If the diffusion process is assumed to be Gaussian, the deformation of Equation (2) condenses to the known diffusion equation in the DTI: (3)$$S_{\Delta }\left(\vec{q}\right)=S_{0}e^{-bg^{T}Dg}$$
where $\vec{g}$ is the unit vector $\vec{q}/\left|\vec{q}\right|$, $b=\tau {\left|\vec{q}\right|}^{2}$, and *D* is the three-dimensional diffusion tensor. DTI data are usually collected using an optimized 30 direction and a constant b-value equal to 1000 s/mm^2^, and the scanning time is relatively short (approximately 6 min). 

Model-free dMRI techniques, like HARDI and QBI, require more gradient directions, such as 82, 120, 257, and 515 [20]. The b-value is generally higher (2000~3000 s/mm^2^) than DTI, and the data scanning time lasts from 15 to 30 min. 

Currently, the most complex sampling scheme is the DSI, which collects the data points of the whole q-space to depict the diffusional behavior of water molecules. Generally, the gradient directions range from 102 to 515, the b_max_ value ranges from 6000 to 12,000 s/mm^2^ [21], and the scanning time is >35 min.

Practically, to exclude the phase shifts arising from tissue motion [22], the probability density function (PDF) is reconstructed by taking the FT of the modulus of the complex MR signal:(4)$${\vec{p}}_{\Delta }\left(\vec{r}\right)=S_{0}^{-1}{\left(2\Pi \right)}^{-3}\int^{\;}\left|S_{\Delta }\left(\vec{q}\right)\right|e^{i\vec{q}\cdot \vec{r}}d^{3}\vec{q}$$

Since ${\vec{p}}_{\Delta }\left(\vec{r}\right)$ is a measured quantity calculated by FT, we simply refer to it as the diffusion spectrum [23].

DSI sampling is initially acquired by combining the 515 gradient directions and b_max_ > 10,000 s/mm^2^, which leads to a long scanning time of up to dozens of hours and significantly prevents its clinical extension. The conventional data analysis steps of DSI are as follows [23]:(1)Image denoising;(2)3D discrete Fourier transform to obtain the PDF;(3)Radial integration for the PDF to acquire the ODF;(4)Calculation of the metrics based on the ODF, such as GFA;(5)ODF-based tractography.

For tractography, Wedeen et al. proposed an algorithm to construct fiber tracks, which has been extensively applied to DSI studies [23]. This method uses a streamlined algorithm, or Eulerian integration, and can be modified to accommodate multiple directions per point. 

Actually, the validity of the Fourier relation between E(q) and PDF is questionable, as the essential requirement for this relation is not satisfied in practice [24]. Interestingly, the reconstruction described in Equation (4) still remains valid with noninfinitesimal diffusion-encoding gradients [25]. Moreover, Lin et al. evaluated the error of the DSI using scanning sequences with both short- and long-gradient pulse widths in a rat model [26]. They found that the bias of the primary orientation between the two sequences was approximately 10°, suggesting that the effect of the finite gradient pulse widths on the primary orientation is not critical. Yang et al. investigated the effect of a finite δ on the DW signal measured as a function of the gradient direction through simulation data and experimental models [27]. Their results indicated that a long δ would lead to a boost in the DW signal in the transverse plane of the fiber and stretch out the shape of the measured diffusion profile, which might improve the contrast between DW orientations.

### 2.1. Improvement in the Scanning Scheme

#### 2.1.1. Challenges

The overlong scanning time under the current sampling scheme is the most serious problem, preventing the clinical application of DSI, which may result in a break in the experiment due to the body condition of patients and excessive head motion during scanning. Various methods have been proposed to shorten the acquisition time of DSI data. Considering the geometric symmetry of diffusion anisotropy, several studies created full-spherical DSI data by exploiting hemispherical or subsampled data (see Table 1). Both phantom and rat brain models showed that this scheme could decrease the DSI acquisition time while preserving the patterns and orientations of PDF [26,28]. However, the cross-term between diffusion and image gradients causes the inaccuracy of hemispherical data for ODF reconstruction [29]. Therefore, it is necessary to correct the cross-term in the post-processing of hemispherical sampling data. 

#### 2.1.2. Technical Advancements

By optimizing the b_max_-value, Kuo et al. found that the sampling scheme with the optimum b_max_-value (6500 for DSI-515 and 4000 for DSI-203 on the 3T scanner) not only effectively decreased the scanning time but also yielded comparable angular precision and accuracy to high sampling schemes [30]. From the sequence perspective, Reese et al. proposed a highly efficient multiplexing method called simultaneous image refocusing (SIR), which modified the usual EPI acquisition using two windowed sinc(t) excitation RF pulses with different frequency offsets, and the results suggested that the modulated sequence reduced the total scan time by nearly one-half [31]. Moreover, two high-efficacy sampling schemes, respectively, using the reduced-encoding DSI (RE-DSI), based on the assumption of the bi-Gaussian diffusion signal curve [32], and the body-centered-cubic (BCC), based on sampling the signal in the center of each unit cell [33], were proposed and were demonstrated to decrease the scanning time of DSI while maintaining the precision and accuracy of ODF. In compressed sensing (CS) applications to accelerate DSI data acquisition, Paquette et al. manifested that when the acceleration factor R = 4, the undersampling data of DSI-128 and DSI-515 could reduce the scanning time to 6 min and 26 min, respectively [34]. Meanwhile, essential information on diffusion properties, such as the ODF, diffusion coefficient, and kurtosis, was preserved. A recent study demonstrated that CS-DSI performed comparably to 3-shell HARDI in the estimation of diffusion and microstructural parameters, and it was a well-suited imaging protocol for dMRI within the scope of a scan-time-limited, high-throughput, and long-term population study [35]. Another recent CS-DSI study presented a comparison of basic functions and q-space sampling schemes for robust CS reconstruction accelerating DSI. They found that Fourier-based CS-DSI showed better reconstruction quality of the diffusion signal and propagator-derived parameters than SHORE-based CS-DSI, but the reconstruction of the orientational information was comparable for the two CS-DSI approaches [36]. Recently, validations in post-mortem [37] and living [38] human brains demonstrated that the accuracy and reliability of the CS-DSI in subsampled images were nearly the same as those generated by the full DSI scheme and further illustrated the utility of the CS-DSI for reliably delineating in vivo brain architecture in an acceptable scan time for clinical applications. Therefore, CS-DSI may be the most optimized scheme to resolve the overlong scanning time at present.

### 2.2. Optimization of the Postprocessing Method

#### 2.2.1. Challenges

Over the past twenty years, an increasing number of advanced post-processing approaches for DSI have emerged to enhance the angular resolution of ODFs and further uncover more complex fiber orientations (see the details in Table 1). It is generally assumed that the peaks in the diffusion ODF (dODF) correspond to the direction of the fiber population, but they cannot provide the actual fiber orientations [12]. 

#### 2.2.2. Technical Advancements

Some studies assume that all fiber bundles in the brain’s white matter share identical diffusion characteristics, thus implicitly assigning any differences in diffusion anisotropy to partial volume effects [39,40]. Then, the DW signal attenuation is expressed as the convolution over the sphere of a response function (the diffusion-weighted attenuation profile for a typical fiber bundle) with a fiber ODF (fODF) [41]. Therefore, the fODF may reflect the real fiber orientations, which can be obtained using spherical deconvolution, and is used to depict the orientation distribution of fiber volume fractions [42,43]. Simulation experiments and human brain studies demonstrated that both the angular resolution of the fODF and the accuracy of fiber tracking based on fODF were better than those of dODF [42,43]. Moreover, other deconvolution methods based on constraint optimization or regularization have also been reported, aiming to handle the negative condition or the background corruption problem [43]. Canales-Rodríguez et al. argued that the FT used to compute the PDF in DSI was based on discrete signals with finite support rather than the whole measurement space. Thus, the PDF obtained from the experiments was the convolution between the true PDF and a point spread function (PSF) [44]. This study demonstrated that after deconvolution, the angular resolution of the ODF was enhanced, and the artifactual peaks and the uncertainty of the local diffusion orientation distribution were reduced. To provide a common deconvolution method for q-space imaging, Yeh et al. proposed a mixed diffusion model in which the fODF was defined as the orientation distribution of the fiber spin density, and the experiments found that the fODF derived from deconvolving the dODF showed consistent fiber orientations regardless of the reconstruction methods and sampling schemes [45]. Yeh et al. further extended the L1 regularization paradigm to dODFs and proposed a diffusion decomposition method to obtain a sparse solution of fODFs and provide a better resolution power for crossing fibers [3]. The subsequent phantom experiment and in vivo study indicated that the angular error of the diffusion decomposition was significantly lower than those of the constrained spherical deconvolution and the ball-and-sticks model, and the fiber orientations resolved by diffusion decomposition were not affected by the different sampling schemes and reconstruction methods [3]. In the DSI implementations, the presence of aliasing due to fast diffusion components like those from pathological tissues can lead to artifactual fiber reconstructions. Lacerda et al. proposed a novel approach including biophysical constraints to compute the ODF, which removes most of these artifacts and offers improved angular resolution [46]. Additionally, a recent study proposed a generalized DSI (GDSI) framework to compute the ensemble average propagator by multiplying the sampling non-uniformity corrected q-space samples with a discrete FT matrix, indicating the GDSI matrix formalism could be used to elucidate the contribution and combination of q-space signals to the dODF [47]. 

## 3. Application

### 3.1. DSI Tractography for White-Matter Fibers

Several tissue structures include a number of myofibers aligned along multiple spatial axes at the microscopic scale. However, the DTI cannot uncover such structural characteristics due to its failure to detect multidirectional fibers. Therefore, one of the important applications of DSI tractography is primarily focused on revealing the complex microstructure of the tissue. For instance, for an anterior slice of the lingual core in bovine tongue, DTI depicted it solely as a region with low anisotropy, whereas DSI revealed two different fiber populations with an explicit orthogonal relationship to each other [48]. Dai et al. found that DSI tractography revealed that the cingulum bundle was less mature when cat myelination was incomplete, whereas DTI tractography tended to terminate in such areas, possibly due to the existence of crossing fibers [49]. By using DSI, Schmahmann et al. identified the major features of 10 long association fiber bundles that matched the observations in the isotope material using autoradiographic histological tract tracing in the monkey brain, whereas the DTI did not observe such precise structural characteristics due to its inability to visualize the crossing fibers and details of the origins, course, and terminations of the white-matter pathways [50]. In human brain studies, Wedeen et al. used DSI to clarify the relationships of adjacency and crossing between cerebral fiber pathways in four nonhuman primate species and humans [51]. They first found that the cerebral fiber pathways formed a rectilinear three-dimensional grid continuous with the three principal axes of development, and cortico-cortical pathways formed parallel sheets of interwoven paths in the longitudinal and medial–lateral axes, in which major pathways were local condensations. Because of the limitation in DTI, the subcomponents and connectivity of the inferior fronto-occipital fasciculus (IFOF) and the superior longitudinal fasciculus (SLF) in human brain are still controversial. The DSI shows high-quality fiber tractography and fewer partial volume effects and false continuation artifacts, and thereby it has been used to reveal more complete connectivity patterns and anatomical details of the IFOF I-V subcomponents [52] and of the SLF I-III subcomponents [53], which are connected to different cortical regions. Similar DSI tractography applications have been reported in recent studies in the tractography of other white-matter pathways, such as the thalamic–prefrontal peduncle [54], pyramidal tracts [55], anterior commissure [56], and corpus callosum [57]. Collectively, these findings suggest a powerful potential of DSI in enhancing tractography for the complexed white-matter fibers.

### 3.2. Cortical Parcellation and Connectivity Reconstruction

The DTI has limited angular resolution and cannot adequately assess the cortical regions. Another application of the DSI in resolving the tissue microstructure is cortical parcellation and connectivity reconstruction. Recent studies used the DSI to segment the ventral [58] and dorsal premotor areas [59] (VPM/DPM); they found that the VPM consists of four subregions, 6v, 4, 3b, and 3a, and the DPM is divided into three areas, 6a, 6d, and 6v. These brain regions showed consistent inter-hemispheric connection but different intra-hemispheric connection patterns. Based on DSI tractography, two recent studies characterized the connections of the middle frontal gyrus (MFG) and inferior temporal gyrus (ITG) to other cortical areas, respectively. The MFG included two major connections of the superior longitudinal fasciculus (which connected the MFG to parts of the inferior parietal lobule, posterior temporal lobe, and lateral occipital cortex) and the inferior fronto-occipital fasciculus (which connected the MFG to the lingual gyrus and cuneus) [60]. The ITG is connected to five major fibers: the U-fiber, the inferior longitudinal fasciculus, the vertical occipital fasciculus, the arcuate fasciculus, and the uncinate fasciculus [61]. Furthermore, a recent study used DSI tractography to organize “pyramid-shaped crossings” of converged U-fibers, which are key anatomical structures to construct the neural network for intricate communications throughout the entire cerebrum [62]. Using the same method, another study delineated the decussating dentato-rubro-thalamic tract, in which the afferent regions were found mainly in the posterior cerebellum, and the efferent fibers were mainly projected to the contralateral frontal cortex, suggesting segregated and parallel cerebellar outputs to cerebral regions [63] (see Table 2).

### 3.3. Clinical Applications 

With the improvement in DSI data acquisition, more attention has recently been paid to its application in clinical diseases, including attention-deficit/hyperactivity disorder (ADHD) [2,64,65], schizophrenia [66,67,68], stroke [69,70,71,72], Parkinson’s disease [73,74], hypertension [75], autism [76], epilepsy [77], and gliomas [78] (see Table 2). 

#### 3.3.1. Disease Diagnosis

The quantitative diffusion scalars of the DSI, especially the track density imaging (TDI) of the crura of fornix (FORX) and the parahippocampal radiation of the cingulum (PHCR), are sensitive enough to define the ipsilateral side for epilepsy patients, with a sensitivity of 89.5% and specificity of 100.0% for PHCR_TDI (AUC = 0.93), and a sensitivity of 95.0% and specificity of 100% for FORX_TDI (AUC = 0.95) [77]. The DSI-based quantitative anisotropy (QA) values of corticospinal tracts (CSTs) in patients with idiopathic normal pressure hydrocephalus (iNPH) were significantly lower than those in healthy controls (HCs), but no significant differences were found between iNPH patients and HCs in the DTI-based FA values, suggesting the DSI may provide more information that can improve the present understanding of the disease mechanism [79]. Another study found that the QA value was correlated with the neuronal diameter/density in the cortical layer IIIc, and its asymmetry showed an overall favorable accuracy (sensitivity = 90.9%, specificity = 89.5%, AUC = 0.96) in the diagnostic testing of hippocampal sclerosis patients [80]. The DSI parameters also showed a good performance, with an accuracy of 83%, sensitivity of 78%, and specificity of 86% in discriminating patients with mild and severe visual defects [81]. 

#### 3.3.2. Progression Prediction 

A prior study demonstrated that the DSI could be helpful for the preoperative prediction of human epidermal growth factor receptor 2 (HER2) in patients with breast cancer, with the finding that the AUC values of the DSI quantitative parameters (range from 0.67 to 0.72) were higher than that of the DTI metric apparent diffusion coefficient (AUC = 0.57) [82]. Another study found that the anatomic integrity of the pyramidal tract (PT) with DSI tractography effectively predicted the postoperative motor function after hemispherectomy; they reported that the AUC of the DSI tractography was 0.84, and the cutoff value of the PT asymmetric ratio was 11.5%, with 100% sensitivity and 75% specificity [83]. The DSI-derived GFA in the ipsilateral medial geniculate body was related to prognosis (sensitivity = 64.7%; specificity = 85.7%; and AUC = 0.80) in patients with unilateral idiopathic sudden sensorineural hearing loss, indicating the GFA value of the ipsilateral medial geniculate body may help to predict recovery outcomes [84]. Another study reconstructed a local connectome matrix from DSI data in patients with aphasia after stroke, and their findings challenged dual-stream accounts that denied a role for the arcuate fasciculus in semantic processing and ventral-stream pathways in language production and illuminated limbic contributions to both semantic and phonological processing for word production [69]. A recent DSI study demonstrated for the first time that distinct aspects of the cortical structural reserve enable basal and complex motor control after stroke. In particular, the recovery of basal motor control may be supported via an alternative route through contralesional M1 and non-crossing fibers of the contralesional CST [72]. Taken together, these findings suggest the DSI is a very potential and powerful technology in studying the mechanism, diagnosis, and progressive prediction of clinical diseases. 

**Table 2 brainsci-13-01497-t002:** A summary of main progresses in DSI applications.

First Author (Ref. #)	Type	Subject	Main Findings
Lacerda et al., 2016 [46]	Methodology	Optimization of postprocessing method	This study proposed a new way of including biophysical constraints to compute the ODF, which removed most of the artifacts due to fast diffusion components like those from pathological tissues and offered improved angular resolution.
Tian et al., 2019 [47]	Methodology	Optimization of postprocessing method	This study proposed a generalized DSI framework to compute the ensemble average propagator, which could be used to elucidate the contribution and combination of q-space signals to the diffusion ODF.
Gilbert et al., 2006 [48]	Application	DSI tractography for white-matter fibers	Diffusion tensor imaging (DTI) depicted the anterior slice of the lingual core in bovine tongue solely as a region with low anisotropy, whereas DSI revealed two different fiber populations with an explicit orthogonal relationship to each other.
Dai et al., 2016 [49]	Application	DSI tractography for white-matter fibers	The cingulum bundle was less mature when cat myelination was incomplete, whereas the DTI tractography tended to terminate in such areas.
Schmahmann et al., 2007 [50]	Application	DSI tractography for white-matter fibers	This study identified 10 major long association fiber bundles that matched the observations in autoradiographic histological tract tracing in the monkey brain, and such precise structural characteristics were not observed by DTI.
Wedeen et al., 2012 [51]	Application	DSI tractography for white-matter fibers	This study first clarified the relationships of adjacency and crossing between cerebral fiber pathways in four nonhuman primate species and humans.
Wu et al., 2016 [52]; Wang et al., 2016 [53]	Application	DSI tractography for white-matter fibers	The DSI revealed a more complete connectivity pattern and anatomical details of the IFOF I-V subcomponents and of the SLF I-III subcomponents.
Sun et al., 2018 [54]; Suo et al., 2021 [55]; Liu et al., 2022 [56]; Wei et al., 2017 [57]	Application	DSI tractography for white-matter fibers	The DSI identified detailed and completed white-matter pathways, including the thalamic–prefrontal peduncle, pyramidal tracts, anterior commissure, and corpus callosum.
Sheets et al., 2020, 2021 [58,59]	Application	Cortical parcellation	The DSI segmented the ventral premotor area into four subregions of 6v, 4, 3b, and 3a and the dorsal premotor area into three areas of 6a, 6d, and 6v.
Briggs et al., 2021 [60]; Lin et al., 2020 [61]	Application	Cortical connectivity reconstruction	The MFG included two major connections of the superior longitudinal fasciculus and inferior fronto-occipital fasciculus. The ITG connected to five major fibers: the U-fiber, inferior longitudinal fasciculus, vertical occipital fasciculus, arcuate fasciculus, and uncinate fasciculus.
Chiang et al., 2020, 2023 [2,64];Tsai et al., 2021 [65]	Application	Attention deficit and hyperactivity disorder (ADHD)	Participants with ADHD showed more rapid development of generalized fractional anisotropy (GFA) in the frontal tracts and showed higher axial diffusivity values in the perpendicular fasciculus, superior longitudinal fasciculus I, corticospinal tract, and corpus callosum compared to the control group.
Wen et al., 2020 [73]; Papageorgiou et al., 2021 [74]	Application	Parkinson’s disease (PD)	The PD patients showed impaired global efficiency and characteristic path length in the DSI-based connected network, which were associated with executive function and episodic memory.
Wang et al., 2020, 2022 [77,80]; Zhang et al., 2023 [82]	Application	Epilepsy	The AUC of the asymmetric indices of the DSI-derived QA value to the lateralization of epilepsy was 0.96, with 0.91 sensitivity and 0.90 specificity; The AUC of DSI tractography was 0.84, with 100% sensitivity and 75% specificity in discriminating patients with epilepsy from healthy controls.
Ni et al., 2020 [76]	Application	Autism spectrum disorder (ASD)	A higher GFA of the tracts was implicated in memory, attention, sensorimotor processing, and perception associated with less dysregulation in TDC but worse dysregulation in ASD.
Zhang et al., 2021 [79]	Application	Idiopathic normal-pressure hydrocephalus (iNPH)	The DSI-based QA values of corticospinal tracts (CSTs) in patients with Inph were lower than those in healthy controls (HCs), but such differences in DTI-based FA were observed between iNPH patients and HCs.
Liang et al., 2021 [81]	Application	Pituitary adenomas	The DSI parameters also showed a good performance, with an accuracy of 0.83, sensitivity of 0.78, and specificity of 0.86 in discriminating patients with mild and severe visual defects
Mao et al., 2022 [82]	Application	Breast cancer	DSI could be helpful for the preoperative prediction of human epidermal growth factor receptor 2 (HER2) in patients with breast cancer, with the findings that the AUC values of DSI quantitative parameters (0.67~0.72) were higher than those of apparent diffusion coefficient (0.57) from DTI.
Zhang et al., 2021 [84]	Application	Idiopathic sudden sensorineural hearing loss	The DSI-derived GFA in the ipsilateral medial geniculate body was related to the prognosis (sensitivity = 64.7%; specificity = 85.7%; AUC = 0.796) in patients with unilateral idiopathic sudden sensorineural hearing loss.
Paul et al., 2023 [72]	Application	Stroke	This study used DSI to demonstrate for the first time that recovery of basal motor control may be supported via an alternative route through contralesional M1 and non-crossing fibers of the contralesional CST.
Salisbury et al., 2023 [68]	Application	First-episode psychosis	White-matter tracts showing associations between QA from DSI and auditory hallucinations were associated with frontal–parietal–temporal connectivity in the cingulum bundle and in the prefrontal interhemispheric connectivity.

## 4. Limitations and Future Outlooks

Although numerous DSI studies have optimized the data sampling scheme and post-processing analysis and applied them to clinical diseases, several problems still remain unsolved. Firstly, the overlong scanning time is still the dominant factor preventing DSI from expanding to clinical applications; therefore, the sample size of studies involving DSI is relatively small, which may affect the robustness of the results. Secondly, several studies have demonstrated that DSI has the superb ability to detect changes in microstructural integrity [49,50,51]. However, it may be insufficient to comprehensively evaluate the changes in tissue microstructural integrity by a single scalar metric, and the sensitivity of different diseases may be distinct with regard to various metrics. Finally, although the fODF obtained from diffusion deconvolution or decomposition produces a high angular resolution, the most prominent problem in these methods is that the feature function is not unified [85]. This may result in an unreliable precision for the fODF. For instance, by comparing 19 common dMRI approaches, Daducci et al. found that the resolution of DSI-based ODF is satisfactory in revealing crossing fibers with high angles (90° and 60°), but it fails to detect crossing fibers with low angles (45° and 30°) [18]. 

Based on the abovementioned limitations, we propose the following research outlooks. Firstly, in the Methodology section, we discussed several methods to shorten the DSI scanning time. CS-DSI may be the most optimized scheme among them in the future, which were used to undersample DSI data (i.e., hemispherical scheme) and reconstruct diffusional signal to achieve the scanning-time reduction [86]. Apart from the gradient direction and b_max_ value, other parameters possibly affect the scanning time, such as the TR, FOV, and bandwidth, which could be optimized to improve the DSI scan scheme [87]. Secondly, with regard to clinical applications, future studies should combine more metrics, such as ODF- and PDF-based measures and hybrid diffusion imaging [88], to examine the underlying pathomechanism of the diseases. Finally, more advanced postprocessing methods should be conducted to increase the angular resolution of the fODF to enhance the success rate of the DSI in revealing fibers with low angles in the future. 

## 5. Conclusions

In summary, although the data sampling scheme and post-processing method of DSI are imperfect, fiber tracking by DSI shows significant advantages in detecting multidirectional diffusion. Moreover, numerous studies have suggested that DSI is capable of uncovering the neural mechanism underlying disorders, implying its powerful potential value in clinical applications. Concerning the existing problems in the methodology and applications of DSI technology, we provide the following suggestions: (1) compressed sensing in the DSI sampling scheme may be an efficient and promising method for scanning-time reduction; (2) the PDF includes more information than the ODF in the DSI post-processing and should be extended in application studies; and (3) DSI studies in clinical diseases need more samples to confirm the reliability and reproducibility of findings. Collectively, the current DSI application studies are emerging rapidly, but those related to methodology are relatively scarce. In the future, we need to pay more attention to the existing problems of the DSI methodology and be more cautious about the findings concerning the clinical application of DSI. 

## Figures and Tables

**Figure 1 brainsci-13-01497-f001:**
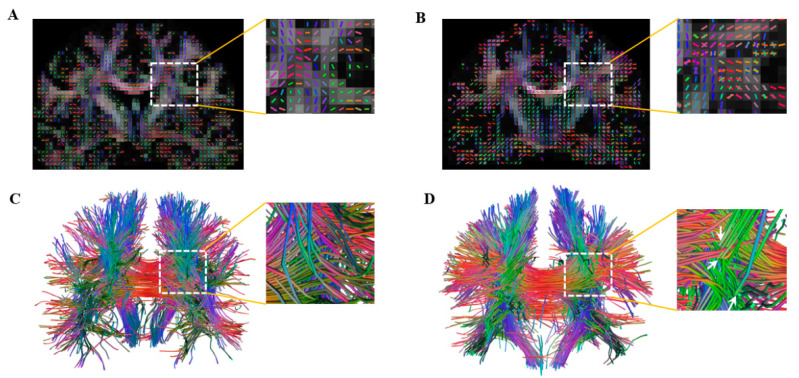
A comparison of DTI (**A**,**C**) and DSI (**B**,**D**) in revealing crossing fiber. (**A**,**B**) the color FA map; (**C**,**D**) the fiber tracts in the whole brain. The white arrows show the crossing fibers in (**D**), whereas they are not revealed in (**C**).

**Figure 2 brainsci-13-01497-f002:**
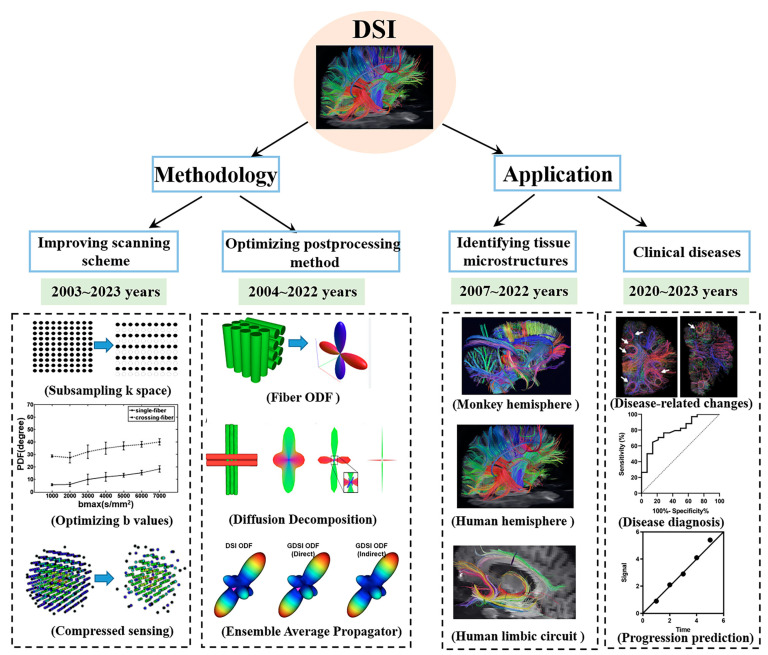
The design of the entire work.

**Table 1 brainsci-13-01497-t001:** A summary of main progresses in DSI methodology.

First Author (Ref. #)	Type	Subject	Main Findings
Lin et al., 2003 [26]; Tefera et al., 2013 [28]	Methodology	Improvement in scanning scheme	Hemispherical or subsampled DSI data decreased the DSI acquisition time while preserving the patterns and orientations of the probability density function (PDF).
Kuo et al., 2008 [30]	Methodology	Improvement in scanning scheme	Optimizing the b_max_-value not only effectively decreased the scanning time but also yielded comparable angular precision and accuracy with high sampling schemes.
Reese et al., 2009 [31]	Methodology	Improvement in scanning scheme	The modulated sequence, which modified the usual EPI acquisition using two windowed sinc(t) excitation RF pulses with different frequency offsets, reduced the total scan time by nearly one-half.
Yeh et al., 2008 [32]; Kuo et al., 2013 [33]	Methodology	Improvement in scanning scheme	Two sampling schemes of reduced-encoding DSI and the body-centered-cubic both decreased the scanning time of DSI while maintaining the precision and accuracy of the orientation distribution function (ODF).
Paquette et al., 2015 [34]; Tobisch et al., 2018, 2019 [35,36]; Jones et al., 2021 [37]; Radhakrishnan et al., 2023 [38]	Methodology	Improvement in scanning scheme	Compressed sensing (CS) accelerated DSI data acquisition while preserving essential information on diffusion properties.
Tournier et al., 2004, 2007, 2008 [39,40,41]; Alimi et al., 2018 [42]; Tsai et al., 2022 [43]	Methodology	Optimization of postprocessing method	They assumed that all fiber bundles in the brain white matter share identical diffusion characteristics and found the fiber ODF might reflect more real fiber orientations than the diffusion ODF.
Canales-Rodríguez et al., 2010 [44]	Methodology	Optimization of postprocessing method	This study argued that the PDF obtained from the experiments was the convolution between the true PDF and a point spread function (PSF). The angular resolution of the ODF was enhanced after deconvolution.
Yeh et al., 2013, 2018 [3,45]	Methodology	Optimization of postprocessing method	The authors proposed a mixed diffusion model and a diffusion decomposition method to obtain a precise solution of fiber ODF. These methods provided a better resolution power for crossing fibers.

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
