# Peer review of "Research Progress in Diffusion Spectrum Imaging"

_brainsci, 2023, doi:10.3390/brainsci13101497_

Round 1
Reviewer 1 Report
1- Please provide references to support the claim made in lines 297-298: "Several studies have demonstrated that DSI has the superb ability to detect changes in microstructural integrity."
2- Consider modifying the font in line 118 for improved readability.
3- Revise the sentence: "Therefore, we recommend that future studies consider incorporating additional metrics, such as ODF- and PDF-based metrics [15], to comprehensively investigate the underlying pathomechanisms of the disease." Does this imply that DSI is a suitable method for exploring the pathomechanisms of diseases?
4- Correct the typographical error in line 251, which appears as "Redundant Crochets."
5- Rewrite section 4.2 from a comparative perspective, highlighting the primary advantages of DSI in contrast to traditional models such as DTI, DKI, and others. Emphasize specific features, metrics, or quantities that DSI can support that other techniques cannot.
6- Consider adding visually appealing figures or diagrams to enhance the paper's presentation and engage the reader.
7- Suggest adding a dedicated section in part 4 (applications) that explores the potential roles of DSI in enhancing tractography results.
Author Response
Q1. Please provide references to support the claim made in lines 297-298: "Several studies have demonstrated that DSI has the superb ability to detect changes in microstructural integrity."
Response: Thanks for your comment. We have added several references to support this claim in lines 350-352 in the revised manuscript.
Q2. Consider modifying the font in line 118 for improved readability.
Response: Thanks for your comment. We have modified the font of equation (4) in line 131 in the revised manuscript.
Q3. Revise the sentence: "Therefore, we recommend that future studies consider incorporating additional metrics, such as ODF- and PDF-based metrics [15], to comprehensively investigate the underlying pathomechanisms of the disease." Does this imply that DSI is a suitable method for exploring the pathomechanisms of diseases?
Response: Thanks for your comment. We have revised this sentence as follows: With regard to clinical applications, future studies should combine more metrics, such as ODF- and PDF-based measures and hybrid diffusion imaging [87], to examine the underlying pathomechanism of the diseases (lines 367-369). Moreover, DSI is a suitable method for exploring the pathomechanisms of diseases, we have provided many examples of DSI clinical applications (lines 306-341).
Q4. Correct the typographical error in line 251, which appears as "Redundant Crochets."
Response: Thanks for your comment. We have deleted the redundant crochets.
Q5. Rewrite section 4.2 from a comparative perspective, highlighting the primary advantages of DSI in contrast to traditional models such as DTI, DKI, and others. Emphasize specific features, metrics, or quantities that DSI can support that other techniques cannot.
Response: Thanks for your comment. We have rewritten this section (termed as 3.3 Clinical diseases in the revised manuscript) according to your valuable suggestions in lines 306-343.
Q6. Consider adding visually appealing figures or diagrams to enhance the paper's presentation and engage the reader.
Response: Thanks for your suggestion. We have added two figures to enhance the paper's presentation in line 78 and 90, respectively.
Q7. Suggest adding a dedicated section in part 4 (applications) that explores the potential roles of DSI in enhancing tractography results.
Response: Thanks for your suggestion. We have added a dedicated section (3.1 DSI tractography for white matter fibers) in Application part to demonstrate the potential roles of DSI in enhancing tractography results (lines 247-278).
Reviewer 2 Report
The abstract lacks clarity and conciseness. It should clearly state the purpose, scope, and key findings of the review. Additionally, it should briefly mention the challenges and advancements in DSI to provide a clear overview to the readers.
1. The paper lacks a clear structure. It should be organized into sections (e.g., Introduction, Methodology, Applications, Progress, Conclusion) to facilitate readability and understanding.
2. The paper mentions that DSI has not been widely used in clinical studies due to scanning time and post-processing issues. However, it fails to explore and discuss potential solutions or recent advancements addressing these limitations. A comprehensive review should cover both advantages and challenges.
3. Citation and References: The paper does not provide citations or references to support its claims and statements. A review paper should be well-referenced to demonstrate the credibility of the information presented.
4. The paper briefly mentions fundamental principles and methodology but lacks depth and detail. It should delve into the technical aspects of DSI, making it useful for both researchers and clinicians.
5. While the paper mentions "important progress of DSI," it does not provide specific examples or detailed analysis of recent advancements, making the claim vague.
6. The abstract contains incomplete sentences, which detracts from the professional quality of the paper. All sentences should be grammatically correct and complete.
7. Correct any grammar or language issues to enhance the overall readability and professionalism of the paper.
8. Explore potential solutions or recent developments that address the limitations of DSI, especially regarding scanning time and post-processing.
9. When discussing progress in DSI, provide specific examples or case studies to illustrate the advancements effectively.
Once these recommendations are implemented, the paper may be suitable for resubmission, and it is encouraged to provide additional value to the field of DSI research.
Improvement is required.
Author Response
Q1. The paper lacks a clear structure. It should be organized into sections (e.g., Introduction, Methodology, Applications, Progress, Conclusion) to facilitate readability and understanding.
Response: Thanks for your comment. We have reorganized the structure of this paper to facilitate readability and understanding.
Q2. The paper mentions that DSI has not been widely used in clinical studies due to scanning time and post-processing issues. However, it fails to explore and discuss potential solutions or recent advancements addressing these limitations. A comprehensive review should cover both advantages and challenges.
Response: Thanks for your comment. We have added both advantages and challenges of DSI concerning its clinical application in lines 188-202, 268-301, 306-341 and 346-371.
Q3. Citation and References: The paper does not provide citations or references to support its claims and statements. A review paper should be well-referenced to demonstrate the credibility of the information presented.
Response: Thanks for your comment. We have carefully checked the paper and provided corresponding references of each statements.
Q4. The paper briefly mentions fundamental principles and methodology but lacks depth and detail. It should delve into the technical aspects of DSI, making it useful for both researchers and clinicians.
Response: Thanks for your comment. We have delved into the technical aspects of DSI in the current paper to make it useful for both researchers and clinicians (see response to Q2 and lines 378-387).
Q5. While the paper mentions "important progress of DSI," it does not provide specific examples or detailed analysis of recent advancements, making the claim vague.
Response: Thanks for your comment. We have provided specific examples of recent advancements of DTI in Table 1 (lines 268-301 and 306-341).
Q6. The abstract contains incomplete sentences, which detracts from the professional quality of the paper. All sentences should be grammatically correct and complete.
Response: Thanks for your comment. We have carefully checked the abstract and revised incomplete sentences and grammatical errors.
Q7. Correct any grammar or language issues to enhance the overall readability and professionalism of the paper.
Response: Thanks for your comment. We have revised the paper according to your suggestions and adopted the MDPI English editing service to improve the language of the manuscript.
Q8. Explore potential solutions or recent developments that address the limitations of DSI, especially regarding scanning time and post-processing.
Response: Thanks for your comment. We have added recent developments addressing the limitations of DSI in lines lines 188-202, 268-301, 306-341 and 346-371.
Q9. When discussing progress in DSI, provide specific examples or case studies to illustrate the advancements effectively.
Response: Thanks for your comment. We have provided specific examples to illustrate the advancements of DTI in Table 1 (lines 268-301 and 306-341).
Reviewer 3 Report
The authors presented the paper "Research progress in diffusion spectrum imaging"
1) The reference list should be improved. You have only seven 2020 year paper and no 2020-2023. The most of the papers are too old. I don't know how you can show the progress in the area without fresh works. The 2020-2023 years papers must be presented. I highly recommend not to use the references older 10 years for all sections, except for historically important works.
2) I highly recommend presenting Figures of MRI images which may show clearly the proposed design of the work. It seems that the section 4 with illustrative examples will be highly improved.
3) I haven't found in the paper any quantitative data (sensitivity, specificity, etc.) about the impovements of DSI throughout the text. Moreover, it will be better to summarize the improvement and applications of the method as Tables.
4) The novelty of the work should be clearly mentioned in Conclusion section and Abstract. Conclusion section is poor. The novelty, limitations, and future outlooks should be clearly mentioned in the Conclusion section.
Minor editing of English language required
Author Response
Q1. The reference list should be improved. You have only seven 2020 year paper and no 2020-2023. The most of the papers are too old. I don't know how you can show the progress in the area without fresh works. The 2020-2023 years papers must be presented. I highly recommend not to use the references older 10 years for all sections, except for historically important works.
Response: Thanks for your comment. We have updated the references by adding recent five years research, and deleted the references older 10 years for all sections, except for few important works.
Q2. I highly recommend presenting Figures of MRI images which may show clearly the proposed design of the work. It seems that the section 4 with illustrative examples will be highly improved.
Response: Thanks for your comment. The design of the work is shown in Figure 2 and specific examples of applications have been illustrated in Table 1 and Table 2
Q3. I haven't found in the paper any quantitative data (sensitivity, specificity, etc.) about the impovements of DSI throughout the text. Moreover, it will be better to summarize the improvement and applications of the method as Tables.
Response: Thanks for your comment. We have summarized the improvements of DSI and added quantitative data concerning methodology and applications in Table 1 (lines 268-301 and 306-341).
Q4. The novelty of the work should be clearly mentioned in Conclusion section and Abstract. Conclusion section is poor. The novelty, limitations, and future outlooks should be clearly mentioned in the Conclusion section.
Response: Thanks for your comment. We have rewritten the Conclusion section and Abstract according to your instructive suggestions (lines 20-27 and 378-387).
Round 2
Reviewer 2 Report
It has undergone one stage of revision and appears to be well-structured and focused. However, there are some minor revisions and corrections that can enhance the clarity and readability of the paper.
Minor revisions:
1. In the abstract, explicitly state the paper's contributions and the main findings from the comprehensive review. This will give readers a clear understanding of the paper's significance.
2, When discussing the results of previous DSI studies, consider categorizing them into different subtopics or themes to enhance readability. For example, you can group studies related to clinical applications, technical advancements, and challenges separately. Provide clear references to specific studies when discussing their findings or contributions.
3.In the section on potential solutions and future directions, provide more details on how optimizing the b-value, subsampling, and using compressed sensing can address the challenges of DSI in clinical settings. Explain the benefits of these approaches concisely.Consider adding a subsection that discusses the potential impact of these solutions on the field of neuroimaging.
Fine
Author Response
Q1. In the abstract, explicitly state the paper's contributions and the main findings from the comprehensive review. This will give readers a clear understanding of the paper's significance.
Response: Thanks for your comment. We have revised the abstract according to your suggestion (lines 25-32).
Q2. When discussing the results of previous DSI studies, consider categorizing them into different subtopics or themes to enhance readability. For example, you can group studies related to clinical applications, technical advancements, and challenges separately. Provide clear references to specific studies when discussing their findings or contributions.
Response: Thanks for your comment. We have added more subtopics in section 2 and section 3 to enhance readability and clear references have been provided to each study mentioned in these sections.
Q3. In the section on potential solutions and future directions, provide more details on how optimizing the b-value, subsampling, and using compressed sensing can address the challenges of DSI in clinical settings. Explain the benefits of these approaches concisely. Consider adding a subsection that discusses the potential impact of these solutions on the field of neuroimaging.
Response: Thanks for your comment. Based on the review of previous studies, we proposed several potential solutions and future directions in section 4 (paragraph 2). These solutions were also summarized in the Abstract with details and corresponding benefits (lines 25-31). The prior inaccurate description “such as optimizing the b-value, subsampling, and using compressed sensing” has been deleted.
Reviewer 3 Report
Thank you for the revised paper.
Minor editing of English language required
Author Response
Thanks for your comment and we have adopted MDPI English Editing service to improve the language of the manuscript.